# How the Mind Creates the Body and What Can Go Wrong: Case Studies of Misperceptions of the Body

**DOI:** 10.3390/healthcare11152144

**Published:** 2023-07-27

**Authors:** Erich Kasten, Jill Julia Eilers

**Affiliations:** Department of Psychology, Faculty of Human Sciences, Medical School Hamburg, 20457 Hamburg, Germany; jill.eilers@googlemail.de

**Keywords:** misperceptions of the body, coenaesthesia, paraesthesia, tactile hallucinations, out-of-body experiences, near-death studies, sexual hallucinations

## Abstract

The review brings together a wealth of case studies, both from the authors’ patients and from the literature, about people whose bodies do not feel as they really should. Body parts suddenly become longer or shorter, heavier or lighter and there may be a loss of body control to the point where individuals feel as if they no longer have a body at all. The article differentiates by type of causes: mental disorders (e.g., psychosis), the influence of drugs on body perception and neurological causes. Depending on the type of body change, examples are given from the categories of sexually toned changes in body perception, out-of-body experiences and near-death experiences. Since there are countless types of body image disorders, the article is limited to a selective selection of the most interesting and sometimes obscure deviations.

## 1. Introduction

As early as 1911, Head and Holmes (1911) were among the first to attempt to understand how the brain maps the human body [1]. They developed one scheme for the passive perception of stimuli on the skin (“superficial scheme”) and another for the position and movement of limbs (“postural scheme”). Most contemporary authors use terms like “body schema”, “body image” or “body representation”. In 1979, Critchley used the term “corporeal awareness” [2]. Here, he tried to integrate an affective and emotional component in addition to the perceptual and conceptual components. As Berlucci and Aglioti noted in 2010 [3], there is still no universally accepted terminology. The present article will therefore not deal intensively with the problem of definitions. This narrative review is more about the fact that there is a large number of disorders that lead to a confusion of the mental picture that our brain draws of its own body. Since there are countless types of body image disorders, the article is limited to selective examples of the most interesting and sometimes obscure deviations. Another conceivable distinction concerns motor and sensory disorders. This article deals primarily with changes in the perception of the body, and motor disorders are only marginally discussed. We will start with a clinical single-case study:

One of our female patients, who suffer from epilepsy and had a brain haemorrhage, described how she once had the feeling that her right hand was far more than one and a half metres away from her body. She had the unreal sensation that her forearm could extend more than a metre. The nonsensical nature of this was completely clear to her, yet the feeling remained. A little later, a massive headache set in, forcing her to go to the hospital. The cause, however, could not be determined there. It was probably a transitory ischaemic attack, i.e., a temporary blood circulatory disturbance in the network of the brain which is responsible for our body perception [4].

The body is the link to our earthly existence and everyone is highly accustomed to feeling and moving this body. The existence of our body, and the fact that we can feel and move it, is the most natural thing in the world. However, the brain first had to learn what actually belongs to the body in a pre- and postnatal area. Little babies kick because they stimulate these areas of the brain and train the perception of what actually happens when they move. 

Every morning when we wake up, the brain forms an image of our own body. One knows that the right foot is part of the body and puts the stocking over it. When you wake up and walk out of the bedroom, you can perform this because a complex network of neuronal assemblies in your head works together. The sensitive nerve fibres play an important role here, constantly reporting which posture each individual limb is currently in; the organ of equilibrium behind the ears informs us about our current position, while our eyes provide us with a visual orientation of the entire room. In the CNS, it is first the somatosensory cortex in the parietal lobe that processes the input from, e.g., Merkel cells, Meissner, Ruffini and Vater-Paccini corpuscles, as well as from the nociceptors, and gives us a sense of what our body is doing. The cerebellum plays a crucial role in keeping the body in a stable position, and the motor cortex in the frontal lobe then controls movements adapted to the situation. The pre-motor cortex and the supplementary motor cortex not only plan complex movements but also contain highly trained sequences of movements that can be largely automated, such as riding a bicycle, swimming or playing a musical instrument. Both areas are on the posterior frontal lobe. The supplementary motor area located in the upper frontal lobe is used for the preparation and execution of actions, i.e., the planning and selection of learned, non-stimulus-induced movements and speech. It triggers voluntary motor activity and is located in front of the premotor cortex, which is used for sensorimotor integration and the preparation and execution of voluntary movements. The network also includes thalamic regions and an area in the transitional region between the temporal and parietal lobes, which has a superordinate hierarchical task and tells us what actually belongs to our own body.

This neuronal network can suffer disturbances, for example, through drugs or temporary circulatory disturbances. However, it can also suffer permanent damage, e.g., in the case of strokes, craniocerebral traumas or other neurological disorders, such as polyneuropathy. One can basically distinguish between two forms of such disorders of bodily sensation. On the one hand, there is the (1) negative variant, in which the affected person feels too little of their body, such as the insensitivity of one half of the body in many patients with paralysis after a stroke. On the other hand, in the (2) positive variant, the affected person feels sensory phenomena that are not present at all. Depending on the symptom, these are called, e.g., tactile hallucinations, paraesthesia and coenaesthesia.

In the case of paraesthesia and tactile hallucinations, the affected person may feel, for example, as if thousands of insects are crawling over them, worms are boring through their skin, they are being pushed from behind or strangers are lying on top of them and crushing them. In alcohol withdrawal delirium, there is often a phase in which sufferers have such tactile hallucinations. Möller [5] reported of a female schizophrenic patient who had the feeling at night that her body was being sawn through and reassembled. However, she only felt pain when the (non-existent) cuts were touched. Another patient, a 50-year-old Spaniard, experienced the tactile hallucination that someone was blowing air into his stomach, as if through a straw, causing his stomach to expand. 

Coenaesthesia and bodily hallucinations are perceptual disorders. According to Fuchs [6], the term “coenaesthesia” is composed of the two Greek words “koiné” and “aísthesis” and originally meant “general sensation”, i.e., the perception of one’s own body condition. As early as 1811, Reil listed a number of disorders under the term “Gemeingefühl” (see Figure 1), which caused the soul to have distorted ideas about the body [7]. Even if these diseases used to have different terms two centuries ago, he probably meant, for example, bulimia, nymphomania, cravings, delusional beliefs about bodily dysfunction or metamorphosis, hysteria and hypochondria. As a result, in the last 200 years, the term “coenaesthesia” has become a component of many mental disorders. Today, the term “coenaesthesia” is found almost exclusively in pathological contexts, such as an abnormal or bizarre bodily sensation in schizophrenia, while the normal recognition of one’s own body is categorised into interoception, exteriorperception and proprioperception (see below).

Coenaesthesia related to bizarre bodily misperceptions and body hallucinations overlap and are sometimes difficult to distinguish. When people hear the term “hallucinations,” many often think of hearing voices or seeing objects or scenes that are not real. However, hallucinations can occur in all sensory modalities; this also includes tactile physical misperceptions.

These types of bodily sensations are often difficult to describe, so that many sufferers feel compelled to use grotesque comparisons. These include, for example, the feeling that individual body parts are suddenly heavy as lead or huge. Albert Hofmann, the inventor of LSD, described such changes in the body schema [8]. Other symptoms can be that the body or individual body parts grow, become thicker, heavier or lighter. Sometimes those affected also have the feeling that they are made of stone, metal, wood or plastic on the inside. Still, other patients suffer from the feeling that their liver is rotting, they feel the heart being cut out, the intestines decomposing, the spleen being parasitised, the pancreas decomposing, the lungs being eaten away or the brain liquefying. Sometimes there are also hallucinations of movement, in which the person has the feeling that body parts are performing movements independently, for which he himself is not at all responsible. In addition, there are pulling and pressure sensations inside the body, even the sensation of being strangled.

Vestibular and kinaesthetic hallucinations change the perception of the position of one’s own body in space. These include, for example, the feeling that everything is spinning, the feeling of endless falling, floating (levitation) or suddenly becoming increasingly heavy and sinking into the bed or the ground.

Such hallucinations can occur with lesions of the upper midbrain and adjacent thalamus [9]. In the waking mode, the thalamus faithfully relays sensory input to the cortex; in the sleep mode, or due to the disturbances of brain function, it does not fulfil this task completely. This change in body recognition involves several neurotransmitters, particularly, acetylcholine and serotonin, which are involved in selective attention and cortical processing. Disorders of acetylcholine and serotonin transmission, which are caused by diseases, medication or drug use, are often accompanied by hallucinations, which can also include bodily illusions, e.g., the feeling of falling into an abyss [9].

## 2. Causes of Bodily Misperceptions

### 2.1. Bodily Misperceptions in Mental Disorders

In the run-up to an illness, when one is unsure whether he or she is still healthy or already sick, many people occasionally experience strange body changes. Physical misperceptions caused by fever, including fever hallucinations, are known to occur. Fever is induced when pro-inflammatory cytokines trigger prostaglandin E2 synthesis by binding to receptors on brain endothelial cells [10]. Fever affects several neurotransmitters, Cox and Lee specifically listed norepinephrine, 5-hydroxytryptamine and acetylcholine, which in turn can severely disrupt the body’s recognition [11].

Delusions of fever, i.e., a form of hallucination during severe infectious diseases, are often described in the literature in children. According to Eggers [12], about 30% of all schizophrenic children suffer from bodily hallucinations or coenaesthesia. Eggers described a 7-year-old girl who experienced the sensation of a snake in her stomach. Additionally, a 9-year-old boy reported feeling stones moving back and forth inside his head. Another child said, “It feels like smoke is moving through my body”. An 11-year-old boy said that he felt his head getting longer and bigger in size.

Changes in body perception are most often described in psychoses, primarily schizophrenia. Here is an example: The patient had been raised in a home where her father had sexually abused and impregnated one of her sisters. In her adolescence, she was then raped by a stranger. At 23, she married a man who had sexual intercourse with her up to five times a day. She then developed delusions and hallucinations and had to be hospitalised several times. During the exploration, she also reported seeing her deceased husband: “When I sleep at night, I suddenly wake up because I feel a stitch on my upper arm or leg. Then, when I wake up, there is a man next to my bed”. The patient stated that her husband stood very still about 1 metre away from the bed and said nothing. She always recognised his face exactly. He wore a long coat that reached down to the floor so that she could not see his feet. She had often spoken to him and called his name because he stuck her in the leg or arm with a needle. But he did not give any answer and always disappeared again abruptly [5].

What is astonishing about schizophrenia is that the sufferers, despite the severity of the symptoms, usually have hardly any insight into the illness. Often, they consider themselves to be completely mentally healthy and are absolutely convinced that all their problems are caused by others. 

Another example was written to us by a 17-year-old student: “I see, feel, hear and sometimes smell things that are not there and this is all day, every second and this is not an exaggeration. I know they are hallucinations because they are not real, but I perceive them at any time I am conscious. I can’t stop it. To give an example: I see skeletons walking around, looking at me and even walking through me. I feel that too, I imagine it. It’s like a third eye with which I perceive these hallucinations, I just see it. Or I can feel it very clearly in my body, even now at this moment. I can still distinguish very well what is real and what is not. Nevertheless, these hallucinations, especially when they started and became ingrained in me, have been extremely hard on me. It is difficult for me to concentrate on something, like reading a book, because these hallucinations are always there. When I read a book, they become even stronger and more intense. It takes an enormous amount of strength for me to still manage everyday life with school in this way”. Having the feeling that a skeleton is wandering through one’s own body is not likely to be very pleasant, but there are clearly worse forms of body perception in psychoses.

Another example from the pool of our personal experiences tells of M., a young man who was not averse to drugs, but in whom a drug-induced psychosis was ultimately triggered: M. called the police. When they arrived at his house, they found 15 g of hashish lying in the middle of the table. M. did not care at all that his entire cannabis stash was to be confiscated. He shouted at the police officers that they should first note down the damage caused by the insects. After the police officers confirmed that there were no insects present, they advised him to lie down in his bed for a while and assured him that tomorrow the reality would look different. M. then ran himself a hot bath to relax. The result was that he discovered a large tapeworm crawling out of his anus in the bath. He screamed like mad and ran naked, dripping with water, out of his flat, down the corridor and out of the house onto the open street, where he shouted warningly at people, “The insects are coming!”; a short time later, he found himself in an emergency ambulance, which again took him to the psychiatric ward.

Möller described in detail the fate of a 19-year-old girl who suffered from catatonic schizophrenia [5]. She herself was a premature baby. Her father died when she was still young, and her mother had since remarried for the third time. As a child, she was weak, unsociable and plagued by severe feelings of inferiority. Later, she worked in the garment industry, but always thought she heard her colleagues making fun of her. She reported having sexual intercourse with her cousin, which later weighed heavily on her. Then she became obsessed with the feeling that she was being blamed by television and radio. One day, according to the file, she suddenly became completely stiff, her head fell to the side and her shoulders and eye area were completely cramped. Afterwards, she believed she was possessed by the devil and thought they wanted to burn her. Epilepsy and other neurological diseases could be ruled out, and she was classified as a catatonia-schizophrenic. In the clinic, she was very sexually uninhibited and asked anyone who came towards her to touch her genitals. She alternately believed that she was the Archangel Gabriel and then again that she had to go to hell. Finally, she was convinced that she was having a child and even felt its movements in her belly. At the same time, she saw angels and devils flying through the air. Once she threw a towel out of the window, convinced it was a snake that had bitten her. The Pope also appeared to her, wearing a long brown robe, a red bishop’s mitre and holding a white book in his hand from which he seemed to be reading. Only his face was somewhat blurred, so that she did not know whether he was alive.

Schizophrenia is a typical illness, but false sensations of the body also occur in many other mental illnesses. Anorexia nervosa, too, is ultimately based on a calculation error of the brain, according to today’s assessment. Suchan et al. showed that there are differences in the extrastriate body area between healthy women and those suffering from anorexia nervosa [13]. Interestingly, this difference decreases measurably after therapy. It is assumed that a specific area of the brain provides false information in this case; no matter how thin a person has become, this area stubbornly reports back that one is still too fat.

People who suffer from gender incongruence, i.e., transgender/transidentity, have a different problem. The appearance, especially of the gender-typical areas of the outer body, does not match the mental gender representation. Psychologically, someone is a woman, but the body is that of a man or vice versa. 

Body Integrity Dysphoria or Body Integrity Identity Disorder is another strange disorder from this group of forms. The outer body of these patients is usually completely intact, but they have the urge to want to be disabled. They do feel complete only when they have achieved a specific disability, e.g., the amputation of a leg [14]. 

Another disorder from this group of forms, which we will only briefly mention by name here, is Body Dysmorphic Disorder, in which the persons concerned get carried away with the feeling that certain body parts look ugly; often a surgical change is sought, although the respective body part corresponds to the average in every respect.

### 2.2. Bodily Misperceptions Caused by Drugs

A large number of body hallucinations can be directly caused by drugs. Albert Hofmann, the inventor of LSD, has already been quoted above: “I myself had small, finely formed hands. When I washed them, it happened far away from me, somewhere in the lower right. It was questionable, but completely immaterial, whether they were my hands at all. [...] I once felt like a figure in surrealist paintings whose limbs are not connected to the body, but are just painted next to it.” [8] (see Figure 2).

A young man who had used LSD and cannabis experienced something similar: “At this place we then smoked the weed, and not ten minutes later an effect completely untypical for marijuana began. It was unbelievably strong, far too strong to bear with a clear conscience. I told myself that the effect was only from the weed and that it would stop pretty soon. But then I looked at my palm and the individual fingers bent, became longer and shorter; my whole hand was wobbling. Then, unfortunately, I had to admit to myself that the effect came from the LSD after all. I thought, oh no, not here, not now of all times. I panicked, took a booklet on drugs and opened the page with the LSD description. Techno music was playing in the background, sounding almost like a piano. The lines of the book seemed to me like black and white piano keys, pressed low in rhythm with the music to produce the sound I was hearing, and then came back to me. They became completely detached from the book, so that I finally put it away. It was the same with my mouth when I wanted to speak, my ears, my sense of touch, my legs and my sense of taste, which made it difficult for me to use any of these senses consciously because it seemed complicated to reach a certain one through the tangle of sensory functions in order to be able to act with it. I gave my body an attempt at a command from the subconscious and without actual control over what legs would do, they just went off in the direction indicated without giving any feedback as to what exactly they were doing or to what extent I would then have control over it. Each of these organs only functioned individually and uncontrollably. When I said something, it seemed as if my mouth was saying something and I was just a spectator of it. It was the same with my actions. My legs walked all by themselves. I had no real control over it at all anymore, because the sense of moving my legs was attached to one of the prongs, and was so far away that I had a hard time getting to it”.

Under the effect of drugs, one can also become a completely disembodied being. In the book Lucifer’s Garden of Light by Olaf Kraemer [15], one can read the example of a young man who experimented with a drug that was unknown to him until then. He had placed a few milligrams of Salvia divinorum on an aluminium foil and heated it. He inhaled the rising vapours in a self-experiment, which was at the same time intended to find the dose, and had already come to the conclusion that his experiment would have no effect when he realised that something terrible had obviously happened, something had gone wrong: “In great despair I tried to find my way back to the real world, searching my memories for details about the living room I had just been in. I tried to remember what my body felt like. But the more I searched for a thread that would reconnect me with the world I was familiar with, the more violently the extract tried to show me something else. I found that the reality I wanted to return to did not exist. It was merely an ephemeral dream. I noticed that I had no directed access to a memory of any state other than my present, disembodied one…”.

After inhaling deodorant propellant, a female student wrote the following experience: “A wall full of children’s photos of me, right behind the teacher, I stare at the pictures waiting for someone to laugh at the photos, that’s when they melt into the blackboard and the wall, some still seem to call out to me as they dissipate. Black rags fly from the ceiling, no one seems to see them, graveyard bells ring, I get up, mumble something and leave the room, rather: my body leaves the room. I watch myself do it”.

Another youngster who was keen to experiment once smoked some DMT (dimethyltryptamine) extract by mistake, thinking it was hashish. The hallucinations, he wrote to me, were comparable to LSD, only much stronger: “I lay on the bed and felt completely disembodied, something like the body wasn’t really there anymore. Getting up to walk around the environment was unthinkable. How could I, without a body and especially without legs to carry me? The intoxication only lasted for about an hour. There is not much more to tell of this intoxication than that I was completely incapacitated and absolutely disembodied. I have never heard of a case where a user got up after taking DMT. In all the examples I know of, the users had to lie down and it was only after an hour that they were responsive enough to get up again”.

Cashman reported LSD experiences as early as 1966: “My body melted away in waves, [...] I felt myself flying out into space, without heaviness or restraints, freed to bathe in the blissful glow of heavenly apparition. [...] There was no time, no place, no self. There was only cosmic harmony. [...] For me, the realities of our limited existence were no longer valid. I had seen the ultimate truths, and no others would be able to stand before them. [...] I felt myself flying out into space, without heaviness and without fetters, freed to bathe in the blissful splendour of heavenly appearance.” [16].

### 2.3. Bodily Misperceptions Due to Neurological Damage

In Alexander Lurija’s book The Man With a Shattered World, the brain-injured soldier Zasetzki also reported a wealth of changes in his own body schema: “Sometimes I sit there and suddenly feel that my head is as big as a table, at least as big. But arms and legs and torso have become tiny. It seems strange and ridiculous to myself when I suddenly remember it! I call these phenomena peculiarities of the body! And when I close my eyes I don’t even know where my right leg is, and for some reason it has always seemed (and been felt by me) as if it were somewhere above the shoulders and even above the head.” [17].

Ultimately, all misperceptions have a neurological basis. Logically, processing errors in the sensory cortex come into question first, because this is where the perception and initial processing of all stimuli from the body occur. However, this is by far not the only brain area, more it is a matter of a disturbance in a network that consists of very different parts.

Using fMRI techniques, in 2001, Downing et al. found an area in the right lateral cortex which gave a stronger response when subjects viewed images of human bodies [18]. Downing named this field “extrastriate body area” Four years later, Peelen and Downing found a second body-selective area in the middle fusiform gyrus [19]. This fusiform body area responds selectively to images of human bodies. The extrastriate and the fusiform body area seem to be sensitive to bodily actions expressing emotions, such as anger, disgust, happiness and fear.

In 2010, Berlucci and Aglioti [3] pointed out that “In the nineteenth century, neurological thinking about the means by which the body communicates with the brain emphasized the importance of the concept of coenaesthesia, a mainly unconscious sense of the normal functioning of the body and its organs which emerges to full consciousness only when one is unwell”. Today, this concept is renamed as interoception, which refers to the inner perception of one’s own bodily processes (e.g., hunger, muscular sensations, pain, temperature (fever), thirst and visceral sensations in the guts). It works together with exteroception and proprioception: Exteroception refers to the perception of the environment through sense organs, while proprioception is the unconscious perception of one’s own movement, position, tension, posture and position in space. 

In 2009, Craig pointed out that interoception, proprioception and exteroception feed the brain with information about the condition of the body [20]. The cortical representation is mainly settled in the insula. According to Craig, the insular cortex has sensory inputs (e.g., gustatory, somatosensory, vestibular and visceral) that are integrated across modalities and are closely connected to the anterior cingulate cortex. They form an emotional network with which sensory reception is linked to conscious feelings and motivations [20]. Self-recognition is also attributed to this network in conjunction with the default network system. Craig wrote in 2009 that the anterior insular cortex is responsible for the integration of all bodily feelings and, when disturbed, can result in errors of body belonging, such as hemiplegia with anosognosia, neglect, body integrity dysphoria, autoscopy or out-of-body experiences and “astral travel” [20].

Autoscopy and heautoscopic hallucinations are not necessarily physically noticeable changes, but one sees oneself from the outside. According to Goldenberg, autoscopic phenomena are associated with temporo-occipital rather than parietal lesions. Usually, they are short-lived and often associated with epileptic seizures originating in the temporal lobes [21].

Phantom feelings are another example of how the interaction between the brain and body no longer works. Ramachandran [22] described a motorcyclist who had lost an arm in an accident. He could extend his phantom arm, wave it in the air, touch things and even has the feeling he could grasp objects with it. Phantom sensations do not only occur for lost extremities; Ramachandran described a female patient who after mastectomy felt phantom breasts, and another man who experienced phantom erections following the removal of his penis.

Most amputees have feelings of a phantom limb, i.e., the missing limb is still there. However, it often feels shorter than the original healthy part of the body or feels like it is in a distorted or even painful position. For example, amputees may feel itching or a twitch in the non-existent body part. Some try to stand up with a leg that is no longer there, and others try to grab things with the amputated arm. This is due to the fact that the areas of the brain that were used for creating the sensation of the missing body part remain intact even after an amputation. In the 1980s, Melzack postulated that the experience of the body arises from a network of interconnected neuronal structures which he called the “neuromatrix” [17]. In addition to the primary sensory cortex in the parietal lobe, an influence of the thalamus is discussed here in particular. After an amputation, however, there is a restructuring, since neurons that no longer receive input search for new tasks. Some regions of the thalamus that originally represented the missing limb remain functional, while other thalamic neurons begin to respond to stimulation in other regions of the body [23] In addition, Melzack recognized that many people who were born without definite limbs also had phantom limbs [24].

In 2000, Brugger and co-authors described a woman born without forearms and legs who described vivid phantom sensations [25]. An fMRI study showed that “movements” of the non-existent body parts did not activate primary sensorimotor areas but rather the premotor and parietal cortex. Such findings show that parts of the body that never existed in the child’s development can nevertheless be anchored in the brain. Possibly, the observation of the movements of other people is added, so that these brain areas do not turn to other tasks.

The problem for those affected is that the corresponding parts of the brain claim that the amputated body part is still there, but the eyes show the opposite. The idea of being able to grasp an object with a phantom hand also does not mean that this object is now being felt. Sooner or later, this brings those affected to a realistic perception of their phantom feelings.

Another example of strange changes is Alice in Wonderland syndrome. Here, Bittmann and co-authors [26] describe the statements of a patient as follows: “The people who talked to me sounded like they were talking very fast. I had the feeling of being upside down. I was with my grandmother and on her red sofa. Later I found out that I was not there, but in my own house. (...) In the evening, lying down in front of the television: Visual perception was like looking through the wrong end of binoculars. Everything was pushed far away”.

Typical of the syndrome is a change in proportions; what is close appears distant, whereas what is far away seems close enough to touch. This also applies to the body, e.g., the head, arms or legs are perceived as disproportionately huge, and often the ground under the feet feels soft. In addition, there is often a loss of orientation, as well as fear and the feeling of “going crazy”. Usually, the sense of time also changes. Causes include migraine attacks, brain damage, drugs or febrile illnesses.

Alien hand syndrome, also called alien limb syndrome (as it can affect all limbs), is usually due to the right and left hemispheres of the brain, which normally exchange information via the corpus callosum, having communication problems, resulting in the left hemisphere of the brain no longer knowing what the right is doing. This leads to one hand making independent movements that are often untargeted, interfere with the activity of the other hand or may even strike the owner of that hand.

In the bestseller “The Man Who Mistook His Wife for a Hat”, Oliver Sacks vividly described the behaviour of a sufferer: “When I asked him what happened at night, he told me straightforwardly that he always found a dead, cold, hairy leg in his bed when he woke up at night. He could not explain where the dead leg came from and would therefore try to push it out of bed with his healthy arm. But it would somehow stick to his body and he could not get it off. Every time he managed to push the leg out of the bed, he would fall behind. In his view, it would be a bad joke by the hospital staff, who would put an amputated human leg in his bed night after night.” [27].

Somatoparaphrenia is a disorder, most often neurological, in which patients actively deny that a particular limb is part of their own body. If you bring evidence, then there are pseudo-justifications as to why it cannot be your own body part. Sometimes these symptoms take on delusional proportions, and the arm or leg can be treated like a strange being [28,29]. Somatoparaphenia differs from asomatognosia, in which there is a passive loss of recognition of one’s own body parts. It is usually caused by a lesion in a network that appears to include the temporo-parietal junction, posterior insula, basal ganglia and thalamo-cortical connections, among others. There may be correlations with Capgras syndrome, in which a previously familiar person suddenly appears strange after a temporal lesion. In the case of somatoparaphrenia, a previously familiar part of the body also appears strange and does not belong to oneself. Parallels have also been made to Body Integrity Dysphoria (Body Identity Integrity Disorder, Amputee Identity Disorder); in this condition, those affected feel the need to amputate a part of the body that is also perceived as not belonging to their own body. However, sufferers neither show a serious neurological brain lesion nor do they deny that the affected body part is their own. Likewise, delusional justifications are not usually presented in the case of BID sufferers, but they are rationally aware of the pros and cons of an amputation [14].

Cotard’s syndrome is even more drastic; in this condition, the patient claims to be dead, smelling his decaying flesh and feeling maggots gnawing and crawling all over him. The sufferers prefer to stay in cemeteries, since they are actually already dead; many refuse to eat. Cotard’s syndrome is a rare condition that has not shown clear neurological damage thus far, and it often occurs in cases of severe depression.

## 3. Examples of Categories of Bodily Misperceptions

### 3.1. Sexually Toned Disorders of Body Perception

Already in the chapter on mental causes and through the use of drugs, sexually tinged body changes appeared frequently. A classic example of sexual body image disorders is Roman Polanski’s 1965 film “Repulsion” (starring Catherine Deneuve), in which a schizophrenic woman alone at home suddenly sees gaping cracks in the walls; later, she has the feeling that she is being raped by unknown men. 

A 20-year-old female student had been with a young man during the summer semester but then ended the relationship. In November, she thought she was being followed by the young man; she felt his hugs, and this had been a happy experience for her. Then feelings of sexual interference began, at first only for about half an hour a day, but then it had become permanent. In the last weeks, orgasms occurred; afterwards, she had been able to sleep for a few hours, but then it always started all over again. 

A female patient who had met a married man during a cure, to whom she felt strongly sexually attracted, experienced a clearly sexually tinged hallucination. She longed for intimate contact with him and he had probably tried, but both were repeatedly disturbed. After the cure, he then separated from her. At home, however, this man appeared to her. She saw him when she lay down in bed. He was standing against the wall, hovering a little above the floor. He was wearing a brown pullover and dark trousers, and his hair was slicked back—she recognised his face exactly. She had asked him to come to her, but he had stopped at the wall. He often told her things that irritated her sexually (she did not want to say what exactly). But every time she saw him, she was very sexually aroused and often had the feeling that he was making physical contact with her [5].

### 3.2. Out-of-Body Experiences

Out-of-body experiences represent a distinct type of change in bodily sensations; one no longer feels one’s own body longer-broader-nearer-distant, but one no longer has one. The following experiences were taken from an internet page: “Later at night (while I was asleep): Out-of-body experience. I suddenly became aware that I was outside my body. But at that point I thought it was too troublesome to make it a really meaningful experience (I didn’t really know what now? and couldn’t think of anything concrete) and suddenly I snapped back into my body”. 

Ronald Siegel, an American professor, has studied the subject of hallucinations caused by total deprivation of stimuli very intensively and conducted research on himself [30]. For this purpose, he had placed himself in a body-warm, completely dark saltwater tank (floating pool, Shakti), which practically no longer allowed any sensory sensations and was also completely muffled acoustically. Ronald Siegel saw nothing, heard nothing and floated weightlessly on the surface of the water. After a short time, the first hallucinations appeared: “And so I found myself here in the tank. In a state of warmth and calm, I now had the feeling that I was beginning to separate from my body—I even felt my spirit hovering a little above my physical self”.

Psychologist Susan Blackmore has had such out-of-body experiences herself [31]. She compares them to near-death experiences (NDEs): During her first year at Oxford, Susan Blackmore had an NDE after several hours on the Ouija board while stoned on marijuana. The experience also occurred during a period of her life when sleep deprivation was common for her. She describes herself as having been in ‘a fairly peculiar state of mind’ when she had the NDE. She described travelling down a tunnel of trees towards a light, floating on the ceiling, and observing her body below, seeing a silver cord connecting her floating astral body, floating out of the building and then over England, and finally floating across the Atlantic to New York. After hovering around New York, Blackmore floated back to her room in Oxford where she became very small and entered her body’s toes. Then she grew very big, as big as a planet at first, and then she filled the solar system and finally she became as large as the universe. This expansion of consciousness which fills the universe can be found in many NDEs.

### 3.3. Near-Death Experiences

Indeed, near-death experiences also often include the feeling of having left one’s own body and floating above the scene. One of my patients told me: “In 1945, after a severe shrapnel injury, I had my first hallucination: I was lying in bed in the hospital and saw two fiery steeds in front of my bed. I said to my mother, who was sitting next to the bed: ‘Come on, get in, the horses are already harnessed...’. Then followed a feeling of lightness—like a floating state—there were intense colours and I had no pain. The state was so pleasant for me that I had no fear of death. Coming back into the body I found unpleasant”.

Such accounts by dying people have since given rise to a separate branch of research. Raymond Moody is to be thanked for the classic on this subject, which appeared in 1975 under the title “Life after Life” and shook the foundations of our Western thinking [32]. With meticulous precision, Moody had collected reports from people who were dead in the clinical sense but could then be revived with the modern methods of emergency and intensive care medicine. An astonishing number of individuals were able to recall the condition, and even more astonishing was that there seems to be a consistent sequence that we go through in the process of dying. Scientists in the field of “near-death studies” have collected thousands of reports from resuscitated persons over the last 50 years, describing these experiences. Here, with astonishing consistency, a “tunnel experience” is first described in the course of the passing, followed by scenes from one’s own life that were thought to have been forgotten, and finally, some people report seeing filigree patterns and landscapes. Often an intense white light is described: “I got up and went through the hall to get something to drink, and in the process my appendix must have burst, as they found out later. I got a fainting spell and fell to the floor. I was suddenly overcome by a feeling of floating, of moving out of my body and back into it with my true being, and at the same time I heard wonderful music. I floated down the hallway out the door, onto the porch surrounded by a lattice. It almost seemed as if a cloud, or rather a reddish mist, suddenly rose around me, and then I floated straight through the grating, as if it wasn’t there at all, and on up into this pure, crystal clear light—a brilliant white light. It was beautiful and so bright, so radiant, but it didn’t hurt the eyes. You can’t describe a light like that at all here on earth. I didn’t really see the light as a person, but it undoubtedly has a personal individuality. It is a light of the highest understanding and perfect love.” [32].

Moody’s paper described prototypical near-death experiences with a specific order; however, more recent studies show that this row sequence occurs only extremely rarely. Most of those affected report only individual parts of these experiences, and only rarely was the whole process experienced and then often not in the sequence prescribed by Moody. There are also cultural differences. It is scientifically disputed whether one really looks over the border at the end of life or whether it is a question of hallucinations as a result of a neuronal catastrophe reaction [33]. 

As early as 1955, Wilder Penfield had shown that the impression of leaving one’s own body can be evoked by stimulating certain regions in the temporal lobe [34]. Our brain contains information about space and time in the right temporal lobe and about the boundaries of our own body in the left. This orientation-association area always makes it clear to us where the limits of our body lie, but it constantly needs information from the sensory organs in order to be able to carry out its work successfully. If it lacks input from outside, the area is no longer able to define the boundaries between the self and the world. The feeling arises of dissolving into something larger. If this area is also undersupplied on the right side, our brain cannot establish a relationship to space and time. The person has a feeling of eternity and infinity. The hippocampus regulates the neuronal flow of information between the different areas of the brain. Newberg conducted experiments with meditation and with Christian nuns. There was a significant difference in the SPECT: At the lowest point of their immersion, no more activity could be measured in the orientation area.

Detachment from one’s own body can be achieved experimentally using a psychotropic drug, ketamine. Under ketamine (an NMDA glutamate receptor antagonist), which was originally developed as an anaesthetic for animals, test subjects lose all sense of their own bodies. They are awake, but have the feeling that their body does not belong to them and that they themselves are only an external observer; unfortunately—as in sleep—there is usually no memory of what happened afterwards, so it is difficult to conduct research in this area. 

Ronald Siegel reported in his book on the effects of involuntary ketamine ingestion in the context of attempted sexual abuse by knock-out drugs: “The ketamine burned when it finally shot into her, causing her to retch and cough. She did not lose consciousness, however. Instead, her head flew away from her body. From some distance, from some unreal place, she watched what was happening. [...] Donna saw the world as if through the upside-down end of a telescope. It was a kind of Lilliputian version. Everything seemed small and distant. She had no sense of time. People were talking, but there were no clear sounds, just a strange buzzing in Donna’s ears.” [30].

Ketamine paralyses the entire motor and sensory system, so the area of the brain that calculates our body position in space no longer receives sufficient data input. As a result, it can apparently happen that there is now a miscalculation. The corresponding brain area arrives at a result that is several metres off our true position and the brain then develops the impression that we can observe ourselves from outside.

In a single-case study by Swiss scientists, these out-of-body experiences could be triggered by electrical stimulation of the cerebral cortex. Professor Blanke and his colleagues found out during the examination of a woman suffering from epilepsy that out-of-body experiences could be triggered in this patient through the electrical stimulation of a part in the posterior temporal lobe of the brain, the so-called gyrus angularis [35]. The angular gyrus is a processing centre for the perception of one’s own body, in which information from all the systems involved (feeling, balance and seeing) is combined. This study also ultimately comes to the theory that out-of-body experiences are attributed to computational errors of the brain. 

The woman was conscious during the procedure and was therefore able to communicate with the doctors. At the first mild stimulation of the angular gyrus at 2–3 mA, the woman reported mild perceptual changes—she felt as if she was falling from a great height or being pulled back into the cushions. At 3.5 mA, the patient suddenly had the sensation of being outside of her body but could only see her legs and abdomen. Two further trials produced the same result, accompanied by a feeling of lightness and flying just beneath the covers. During these trials, the woman had a feeling as if her legs were coming towards her face at an increasingly rapid pace, making evasive movements. When she lifted her arm, she thought it would come towards her and hit her. According to Blanke and his colleagues, out-of-body experiences are seen as a failure of our brain to reconcile the complex sensations of our body with incoming information.

During an out-of-body experience (OBE), a person appears to be awake and perceives their body and the world from a place outside their own body. Sufferers report something like the following: “I was lying in bed and was about to fall asleep when I was overcome by the clear impression that I was at ceiling height and looking down at my body in bed. I was very startled and frightened; immediately afterwards I felt that I was consciously back in my body on the bed”. 

Based on recent findings, a cognitive model has been proposed according to which OBEs are related to a lack of integration of proprioceptive, tactile and visual information of a person’s body. This can lead to seeing one’s body in a position that does not correspond to the actual perceived position of one’s body. This is often compounded by vestibular dysfunction. 

## 4. Conclusions

Owning the body and responding to our will seem perfectly normal in everyday life—until damage occurs. Changes in body awareness occur in many disorders, and they can be caused by drugs or medication (e.g., ketamine), psychiatric and neurological diseases, out-of-body experiences and especially, in the context of near-death experiences. Awareness of the body is not only dependent on the somatosensory cortex. Recent studies show a complex network of nerve cells that connect from the parietal lobe to the frontal lobe, the insular cortex and the cerebellum.

Moro and co-authors [36] tested the hypothesis that disturbances in the sense of limb ownership are associated not only with discrete cortical lesions but also with disconnections of white-matter tracts supporting specific functional networks. These authors investigated a group of 49 patients with right-hemisphere damage (23 with and 26 without limb disownership). The results revealed that disturbances in the sense of ownership are associated with lesions in the supramarginal gyrus and disconnections of a fronto-insular-parietal network, involving the frontal-insular and frontal inferior longitudinal tracts. These findings lead to propose that the sense of body ownership involves the convergence of bottom-up, multisensory integration and top-down monitoring of sensory salience based on contextual demands.

Even if we now understand the nerve connections in neural networks better, it remains a mystery how exactly the knowledge about our body image arises. As Gabourey Sidibe in an internet page says “It doesn’t have anything to do with how the world perceives you. What matters is what you see”.

## Figures and Tables

**Figure 1 healthcare-11-02144-f001:**
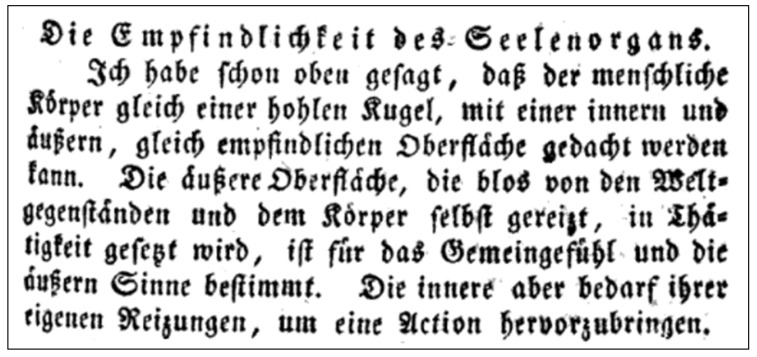
In the text written in 1811 by Johann Christian Reil, he compares the body to a hollow sphere; the outer surface is stimulated by the outside world and is responsible for the “common feeling” (i.e., coenaesthesia), and the inside of the sphere performs its own stimulation: “The sensitivity of the soul organ. I have already said above that the human body can be thought of as a hollow sphere, with an inner and outer surface of equal sensitivity. The outer surface, set in motion merely by the objects of the world and the body itself, is intended for the common feeling and the outer senses. But the inner needs its own stimuli in order to produce an action”.

**Figure 2 healthcare-11-02144-f002:**
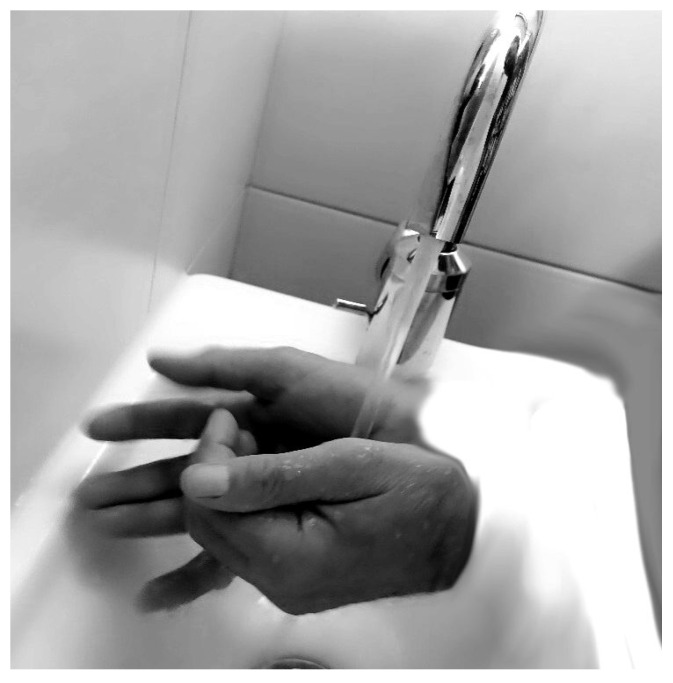
Albert Hofmann described how, when he washed his hands under the influence of LSD, he felt that those parts of his body did not belong to him.

## Data Availability

Not applicable.

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
