# Peer review of "How the Mind Creates the Body and What Can Go Wrong: Case Studies of Misperceptions of the Body"

_healthcare, 2023, doi:10.3390/healthcare11152144_

Round 1

Reviewer 1 Report

This essay presents an entertaining enumeration of misperceptions and distorted experiences of one’s own body. It is OK as a review, even if it is highly selective. For laypeople, the number of different disturbances mentioned may seem impressive. It represents, however, only a tiny selection of alterations of corporeal awareness as they are reported in health and disease. The problem is not the lack of exhaustiveness; the authors could explicitly state that they write about highly selected instances. The major problem is the lack of a conceptual binding that allows some insights into how the body is shaped by the mind as mediated by the brain. To assist in improvig the general presentation of the authos’ list of loosely connected disorders, I may first point out some delicate issues, which could (and should…) be clarified in a re-worked version of the essay. I then list some no-goes, which MUST be corrected before the essay can be considered for publication in «Healthcare» (or elsewhere).

SOME DELICATE ISSUES:

11)     The term «coenaesthesia» pops already out in the (sub)title of the MS. It is welcome, as most contemporary authors use terms like «body schema» or «body image» and think that everybody will uniformly understand what is meant by these terms (Critchley used «corporeal awareness» to avoid confusion). In fact, however, they have much muddled the rich literature on body representation, both clinically and experimentally, which may be briefly and critically discussed by the authors. While the term c. is used appropriately in the title, the definition is misleading on p. 2, line 57 («… are called tactile hallucinations, paraesthesia or coenaesthesia»). Here, c. is defined as a disorder – which is clearly wrong, and probably not intended by the authors. (Cf. also on p.2, line 69 «Coenaesthesia or bodily hallucinations are also physical perceptual disorders.») I recommend to provide a reference to the history of the term «coenaesthesia», from its origin in the late 18th century to the present days: Thomas Fuchs: Z Klin Psychol Psychopathol Psychother 1995;43(2):103-12.

22)     Use and discussion of the term c. could profit from placing it into the larger group of interoceptive disorders; in fact, the absence of the term «interoception» in work about bodily experiences and illusions cannot be really understood (Reil, who coined the term is known today as THE forerunner of interoception and for the «island of Reil», referred to by the authors in the section «Conclusion» as the insular cortex – the insula could have received more attention, in this referee’s opinion).

33)     That the sign of somatoparaphrenia is not discussed, is not intelligible. And given the first author’s expertise in body integrity disorder, I missed some more notes (in addition to the brief mentioning on p. 4) in that direction (authorities have erroneously overstressed the phenomenal similarity between somatoparaphrenia and the amputation variant of BID – this could be clarified). A discussion of BID would also nicely fit in the section on body disorders with a sexual connotation.

44)     Having mentioned this section, I note the weakness associated with it. «Sexually toned» disorders are discussed as if they would constitute a class of bodily misperceptions that would deserve an own conceptual status. This is of course not the case. Mentally, neurologically, drug-induced etc. bodily disorders can have a sexual component; there is neither a hierarchy in the authors chapter titels, nor is there any logic behind the restriction to the type of disorders selected. Also, in the Abstract, a classification according to the rubriques «mental» and «neurological» is suggested, which is obviously problematic for an educated essay on personal reports of bodily experiences.

55)     Some shortcomings in introduced concepts should be removed. For instance, in view of the authors’ repeated emphasis that it remains unclear how we move our limbs or how we ultimately come to know what it means to have a body, it is most puzzling that terms like volition, agency, intentional binding and somatognosia are avoided, throughout. If the authors are critical towards these terms, they may here have a place for being so. But writing about movement initiation and the intention to move without using the keyword «agency» smacks of an absence of knowledge of the respective literatures (plural intended!). In this context, I would suggest that an initial conceptual classification of bodily disorders into motor vs. sensory would make sense. Such distiction is alluded to between the authors’ lines, but it is not made explicit.

6) The topic of phantom sensations is briefly touched upon. The trivial comment is made, that not only limbs can have a phantom existence (but also breasts and penises – these examples, btw, would provide a link to the sexually toned disorders, if only discussed in more depth). There, reference to a well-known popular researcher in the field is made, but the context of this reference is uncritical and as sensationalistic as the work by this authority himself (e.g., «He could extend his phantom arm, wave it in the air, touch things and even grasp objects with it» p. 6, lines 259-60. This cries out for an explanation: what does it mean that objects can be grasped by a phantom hand? Here, keen observation is mingled with popular nonsense – which is disadvantageous for a review essay like the authors’). Many conceptually interesting questions about phantom sensations are not touched (e.g. what is the role of vision in phantom sensations (and in the experience of having a body more generally)? are there phantom sensations of inner organs? What can we learn from the fact that people born witout a limb can nevertheless experience phantom sensations of that particular missing limb?). Of special importance in connection with the authors’ attempt to present a phenomenology of bodily perceptual disorders would be the concept of phantom body. It would be especially key for the understanding of phenomena like asomatognosia or out-of-body experiences. It might help structure the whole essay and give it some essence beyond a list-like enumeration.

SOME NO-GOES

11)     Some work cited in the text is given wrong titles, probably for the reason that it was translated from the German version (Polanski’s movie is «Repulsion», in English, not «Disgust»). Related to this point is that the work by Lurija, Sacks, Ramachandran and Blakeslee (maybe there are others) MUST be cited in the original, or in English, but not in German (reference list). In the text (p.5), Lurija’s book is given a strange title, an obviously wrong translation from the German version; English titel is «The Man with a Shattered World. The History of a Brain Wound».

22)     Please correct: «the pre-motor cortex (supplementary motor cortex)…» PM cortex and SMA are worlds apart!

33)     Philosophically too naïve formulations should be avoided (last line p.5, first p. 6: «the associated brain part remains and continues to feel the body part» No brain part feels anything; it is the person, who feels) as should too popular and sloppy formulations, as they reduce credibility of the authors’ competences (p.2, line 89, «Bad illnesses seem to be able to mess up the neurotransmitters in the brain…»). On p. 1, lines 24, 25: …in the area of the brain that is responsible for our body perception» Such simplifications lessen the impact of the authors’ intention to show that coenaesthesia is something way more complex than being a matter of just one brain area! Unfortunately, similar lapses are found passim in the essay and should be corrected, as they tend to shatter the authors’ authority.
I think that most such passages could be corrected if the essay is edited by a native English speaking person. Ideally, such a person would have a background in academic psychology. Which leads me to the last point:

These no-goes may in part be due to language problems and make it indispensable that the authors should ask a native English speaking person to correct their language. The nontechnical style of the essay is welcome, but expressions are at times difficult to understand (twice on p. 4 «this group of forms»), especially when nuances in experiential or phenomenal descriptions are important for grasping an intended classification, categorization or comparison with related bodily misperceptions.  

See my comments in the evaluation, above.

Author Response

REVIEW #1

This essay presents an entertaining enumeration of misperceptions and distorted experiences of one’s own body. It is OK as a review, even if it is highly selective. For laypeople, the number of different disturbances mentioned may seem impressive. It represents, however, only a tiny selection of alterations of corporeal awareness as they are reported in health and disease. The problem is not the lack of exhaustiveness; the authors could explicitly state that they write about highly selected instances.

We included in the abstract: Since there are countless types of body image disorders, the article is limited to a selective selection of the most interesting and sometimes obscure deviations.

The major problem is the lack of a conceptual binding that allows some insights into how the body is shaped by the mind as mediated by the brain. To assist in improvig the general presentation of the authos’ list of loosely connected disorders, I may first point out some delicate issues, which could (and should…) be clarified in a re-worked version of the essay. I then list some no-goes, which MUST be corrected before the essay can be considered for publication in «Healthcare» (or elsewhere).

SOME DELICATE ISSUES:

1)     The term «coenaesthesia» pops already out in the (sub)title of the MS. It is welcome, as most contemporary authors use terms like «body schema» or «body image» and think that everybody will uniformly understand what is meant by these terms (Critchley used «corporeal awareness» to avoid confusion). In fact, however, they have much muddled the rich literature on body representation, both clinically and experimentally, which may be briefly and critically discussed by the authors.

Page 1:

As early as 1911 Head and Holmes (1911) were among the first to attempt to understand how the brain maps the human body. They developed one scheme for the passive perception of stimuli on the skin (“superficial scheme”)and another for the position and movement of limbs (“posural scheme”). Most contemporary authors use terms like "body schema", "body image" or "body representation". In 1979 Critchley used the term "corporeal awareness". Here he tried to integrate an affective and emotional component in addition to the perceptual and conceptual component. As Berlucci & Aglioti noted in 2010, there is still no universally accepted terminology. The present article will therefore not deal intensively with the problem of definitions, this narrative review is more about the fact that there is a large number of disorders that lead to a confusion of the mental picture that our brain draws of its own body. Since there are countless types of body image disorders, the article is limited to selective examples of the most interesting and sometimes obscure deviations. We will start with a clinical single case study:

While the term c. is used appropriately in the title, the definition is misleading on p. 2, line 57 («… are called tactile hallucinations, paraesthesia or coenaesthesia»).

Depending on the symptom, these are called e.g. tactile hallucinations, paraesthesia and coenaesthesia.

Here, c. is defined as a disorder – which is clearly wrong, and probably not intended by the authors. (Cf. also on p.2, line 69 «Coenaesthesia or bodily hallucinations are also physical perceptual disorders.») I recommend to provide a reference to the history of the term «coenaesthesia», from its origin in the late 18th century to the present days: Thomas Fuchs: Z Klin Psychol Psychopathol Psychother 1995;43(2):103-12.

According to Fuchs (1995), the term coenesthesia is composed of the two Greek words 'koiné' and 'aísthesis', and originally meant "general sensation", i.e. the perception of one's own body condition. As early as 1811, Reil listed a number of disorders under the term “Gemeingefühl” (see Fig. 1), which gave the soul distorted ideas about the body. Even if these diseases two centuries ago used to have different terms, he probably meant, for example, bulimia, nymphomania, cravings, delusional beliefs about bodily dysfunction or metamorphosis, hysteria and hypochondria. As a result, in the last 200 years coenesthesia has become a component of many mental disorders. Today the term coenesthesia is found almost exclusively in pathological contexts, for example as an abnormal or bizarre bodily sensation in schizophrenia, while the recognition of one’s own body is named as “interoception” (see below).

Fig. 1: Text from Johann Christian Reil (1811) in which he compares the body to a hollow sphere, the outer surface is stimulated by the outside world and is responsible for the “common feeling” (i.e. conaesthesia), the inside of the sphere needs its own stimulation.

Coenaethesia as a term for bizarre bodily misperceptions and body hallucinations overlap and are sometimes difficult to distinguish. When people hear the term "hallucinations," many only think of hearing voices or seeing objects or scenes that are not real. However, hallucinations can occur in all sensory modalities; this also includes tactile physical misperceptions.

2)     Use and discussion of the term c. could profit from placing it into the larger group of interoceptive disorders; in fact, the absence of the term «interoception» in work about bodily experiences and illusions cannot be really understood (Reil, who coined the term is known today as THE forerunner of interoception and for the «island of Reil», referred to by the authors in the section «Conclusion» as the insular cortex – the insula could have received more attention, in this referee’s opinion).

Ultimately, all of these misperceptions have a neurological basis. Logically, processing errors in the sensory cortex come into question first, because this is where the perception and initial processing of all stimuli from the body takes place. However, this is by far not the only brain area, more it is a matter of a disturbance in a network that consists of very different parts.

Using fMRI techniques n 2001 Downing et al. found an area in the right lateral cortex which gave a stronger response when subjects viewed images of human bodies. Downing named this field extrastriate body area. Four years later, Peelen and Downing found a second body-selective area in the middle fusiform gyrus. This fusiform body area responds selectively to images of human bodies. The extrastriate and the fusiform body area seem to be sensitive to bodily actions expressing emotions as e.g. anger, disgust, happiness and fear.

In 2010 Berlucci and Aglioti pointed out, that “In the nineteenth century, neurological thinking about the means by which the body communicates with the brain emphasized the importance of the concept of coenesthesia, a mainly unconscious sense of the normal functioning of the body and its organs which emerges to full consciousness only when one is unwell.” Today this concept is renamed as interoception, which refers to the inner perception of one's own bodily processes (e.g. hunger, muscular sensations, pain, temperature (fever), thirst, and visceral sensations in the guts). It works together with exteroception and proprioception: Exteroception refers to the perception of the environment through sense organs, while proprioception is the unconscious perception of one's own movement, position, tension, posture and position in space.

In 2009 Craig pointed out that interoception, proprioception and exteroception feed the brain with information about the condition of the body. The cortical representation is mainly settled in the insula. According to Craig (2009), the insular cortex has sensory inputs (e.g., gustatory, somatosensory, vestibular, and visceral) that are integrated across modalities and are closely connected to the anterior cingulate cortex. They form an emotional network with which sensory reception is linked to conscious feelings and motivations (Critchley 2005). Self-recognition is also attributed to this network in conjunction with the default network system. Craig wrote in 2009 that the anterior insular cortex is responsible for the integration of all bodily feelings and, when disturbed, can result in errors of body belonging such as hemiplegia with anosognosia, neglect, body integrity dysphoria , autoscopy or out-of-body experciences and "astral travel".

Heautoscopic hallucinations are not necessarily physically noticeable changes, but one sees oneself from the outside. According to Goldenmberg (2002), autoscopic phenomena are associated with temporo-occipital rather than parietal lesions. Usually they are short-lived and often associated with epileptic seizures originating in the temporal lobes.

3)     That the sign of somatoparaphrenia is not discussed, is not intelligible. And given the first author’s expertise in body integrity disorder, I missed some more notes (in addition to the brief mentioning on p. 4) in that direction (authorities have erroneously overstressed the phenomenal similarity between somatoparaphrenia and the amputation variant of BID – this could be clarified). A discussion of BID would also nicely fit in the section on body disorders with a sexual connotation.

Somatoparaphrenia is a disorder, most often neurological, in which patients actively denies that a particular limb is part of their own body. If you bring evidence, then there are pseudo-justifications as to why it can't be your own body part. Sometimes these symptoms take on delusional proportions, and the arm or leg can be treated like a strange being. Somatoparaphenia differs from asomatognosia, in which there is a passive loss of recognition of one's own body parts. It is usually caused by a lesion in a network that appears to include the temporo-parietal junction, posterior insula, basal ganglia, and thalamo-cortical connections, among others. There may be correlations with Capgras syndrome, in which a previously familiar person suddenly appears strange after a temporal lesion. In the case of somatoparaphrenia, a previously familiar part of the body also appears strange and does not belong to oneself. Parallels have also been made to Body Integrity Dysphoria (Body Identity Integrity Disorder, Amputee Identity Disorder); here, those affected feel the need to amputate a part of the body that is also perceived as not belonging to their own body. However, sufferers do not show a serious neurological brain lesion, nor do they deny that the affected body part is their own. Likewise, delusional justifications are not usually presented in the case of BID sufferers, but they are rationally aware of the pros and cons of an amputation (Kasten, 2023).

4)     Having mentioned this section, I note the weakness associated with it. «Sexually toned» disorders are discussed as if they would constitute a class of bodily misperceptions that would deserve an own conceptual status. This is of course not the case. Mentally, neurologically, drug-induced etc. bodily disorders can have a sexual component; there is neither a hierarchy in the authors chapter titels, nor is there any logic behind the restriction to the type of disorders selected. Also, in the Abstract, a classification according to the rubriques «mental» and «neurological» is suggested, which is obviously problematic for an educated essay on personal reports of bodily experiences.

The article differentiates by type of causes: mental disorders (e.g. psychosis), the influence of drugs on body perception, and neurological causes. Depending on the type of body change, examples from the categories sexually toned changes in body perception, out-of-body experiences, and near-death experiences are also given.

Chapters are now:

  1. INTRODUCTION
  2. CAUSES OF BODILY MISPERCEPTIONS

2.1. Bodily misperceptions in mental disorders

2.2. Bodily misperceptions caused by drugs

2.3. Bodily misperceptions due to neurological damage

III. EXAMPLES OF CATEGORIES OF BODILY MISPERCEPTIONS

3.1. Sexually toned disorders of body perception

3.2. Out-of-body experiences

3.3. Near-death experiences

  1. CONCLUSIONS

5)     Some shortcomings in introduced concepts should be removed. For instance, in view of the authors’ repeated emphasis that it remains unclear how we move our limbs or how we ultimately come to know what it means to have a body, it is most puzzling that terms like volition, agency, intentional binding and somatognosia are avoided, throughout. If the authors are critical towards these terms, they may here have a place for being so. But writing about movement initiation and the intention to move without using the keyword «agency» smacks of an absence of knowledge of the respective literatures (plural intended!). In this context, I would suggest that an initial conceptual classification of bodily disorders into motor vs. sensory would make sense. Such distiction is alluded to between the authors’ lines, but it is not made explicit.

Deleted: Basically, you can't really explain how we move our body. You can shout out loud to your right arm to stretch upwards, but nothing will happen; it's only when we "want" to, on some strange level that can't be explained in words, that the movement happens.

Inserted in the Introduction:

Another conceivable distinction concerns motor and sensory disorders. This article deals primarily with changes in the perception of the body, motor disorders are only marginally discussed.

6) The topic of phantom sensations is briefly touched upon. The trivial comment is made, that not only limbs can have a phantom existence (but also breasts and penises – these examples, btw, would provide a link to the sexually toned disorders, if only discussed in more depth). There, reference to a well-known popular researcher in the field is made, but the context of this reference is uncritical and as sensationalistic as the work by this authority himself (e.g., «He could extend his phantom arm, wave it in the air, touch things and even grasp objects with it» p. 6, lines 259-60. This cries out for an explanation: what does it mean that objects can be grasped by a phantom hand? Here, keen observation is mingled with popular nonsense – which is disadvantageous for a review essay like the authors’). Many conceptually interesting questions about phantom sensations are not touched (e.g. what is the role of vision in phantom sensations (and in the experience of having a body more generally)? are there phantom sensations of inner organs? What can we learn from the fact that people born witout a limb can nevertheless experience phantom sensations of that particular missing limb?). Of special importance in connection with the authors’ attempt to present a phenomenology of bodily perceptual disorders would be the concept of phantom body. It would be especially key for the understanding of phenomena like asomatognosia or out-of-body experiences. It might help structure the whole essay and give it some essence beyond a list-like enumeration.

Most amputees have feelings of a phantom limb , i.e. that the missing limb is still there. However, it often feels shorter than the original healthy part of the body, or feels like it is in a distorted or even painful position. For example, amputees may feel itching or a twitch in the non-existent body part. Some try to stand up with a leg that is no longer there, others try to grab things with the amputated arm. This is due to the fact that the areas of the brain that were used to create a feeling the missing body part are still there even after an amputation. In the 1980s, Melzack postulated that the experience of the body arises from a network of interconnected neuronal structures that he called the "neuromatrix". In addition to the primary sensory cortex in the parietal lobe, an influence of the thalamus is discussed here in particular. After an amputation, however, there is a restructuring, since neurons that no longer receive input search for new tasks. Some regions of the thalamus that originally represented the missing limb remain functional, while other thalamic neurons begin to respond to stimulation in other regions of the body (Davis et al., 1998). In addition, Ronald Melzack recognized that many people who were born without definite limbs also had phantom limbs.

In the year 2000 Brugger and co-authors described a woman born without forearms and legs who described vivid phantom sensations. An fMRI study showed that "movements" of the nonexistent body parts did not activate primary sensorimotor areas, but rather the premotor and parietal cortex. Such findings show that parts of the body that never existed in the child's development can nevertheless be anchored in the brain. Possibly the observation of the movements of other people is added, so that these brain areas do not turn to other tasks.

The problem for those affected is that the corresponding parts of the brain claim that the amputated body part is still there, but the eyes show the opposite. The idea of being able to grasp an object with a phantom hand also does not mean that this object is now being felt. Sooner or later, this brings those affected to a realistic perception of their phantom feelings.

SOME NO-GOES

  • Some work cited in the text is given wrong titles, probably for the reason that it was translated from the German version (Polanski’s movie is «Repulsion», in English, not «Disgust»). Related to this point is that the work by Lurija, Sacks, Ramachandran and Blakeslee (maybe there are others) MUST be cited in the original, or in English, but not in German (reference list). In the text (p.5), Lurija’s book is given a strange title, an obviously wrong translation from the German version; English titel is «The Man with a Shattered World. The History of a Brain Wound».

A classic example of sexual body image disorders is Roman Polanski's 1965 film "Repulsion" (starring Catherine Deneuve), in which a schizophrenic woman alone at home suddenly sees gaping cracks in the walls; later she has the feeling she is being raped by unknown men.

In Alexander Lurija's book "The Man With a Shattered World", the brain-injured soldier Zasetzki also reported a wealth of changes in his own body schema

Lurija A. (1987) The Man with a Shattered World. The History of a Brain Wound.  Harvard University Press; Reprint Edition.

Sacks O. (2015) The man who mistook his wife for a hat. Picador; 13. Edition 

  • Please correct: «the pre-motor cortex (supplementary motor cortex)…» PM cortex and SMA are worlds apart!

The pre-motor cortex and the supplementary motor cortex plan complex movements, but also contains highly trained sequences of movements that can then be largely automated, such as riding a bicycle, swimming or playing a musical instrument. Both areas are on the posterior frontal lobe; the supplementary motor area on the upper frontal lobe is used for the preparation and execution of actions, i.e. the planning and selection of learned, non-stimulus-induced movements and speaking. It triggers voluntary motor activity and is located in front of the premotor cortex, which is used for sensorimotor integration and the preparation and execution of voluntary movements.

  • Philosophically too naïve formulations should be avoided (last line p.5, first p. 6: «the associated brain part remains and continues to feel the body part» No brain part feels anything; it is the person, who feels) as should too popular and sloppy formulations, as they reduce credibility of the authors’ competences (p.2, line 89, «Bad illnesses seem to be able to mess up the neurotransmitters in the brain…»). On p. 1, lines 24, 25: …in the area of the brain that is responsible for our body perception» Such simplifications lessen the impact of the authors’ intention to show that coenaesthesia is something way more complex than being a matter of just one brain area! Unfortunately, similar lapses are found passim in the essay and should be corrected, as they tend to shatter the authors’ authority.
    I think that most such passages could be corrected if the essay is edited by a native English speaking person. Ideally, such a person would have a background in academic psychology. Which leads me to the last point:

Deleted: After amputation of a body part, the associated brain part remains and continues to feel the body part

This is due to the fact that the areas of the brain that were used to create a feeling of the missing body part are still there even after an amputation.

Bad illnesses seem to be able to mess up the neurotransmitters in the brain in such a way that a variety of strange symptoms can occur. Fever affects several neurotransmitters, Cox and Lee specifically listed norepinephrine, 5-hydroxytryptamine, and acetylcholine, which in turn can severely disrupt the body's

It was probably a transitory ischaemic attack, i.e. a temporary circulatory disturbance in the network of the brain which is responsible for our body perception (Kasten & Poggel, 2009).

Such hallucinations can occur with lesions of the upper midbrain and adjacent thalamus (Epstein et al., 2002). In waking mode, the thalamus faithfully relays sensory input to the cortex; in sleep mode, or due to disturbances of the brain-function it does not fulfill this task completely. This change of body-recognition involves several neurotransmitters, particularly acetylcholine and serotonin, which are involved in selective attention and cortical processing. Disorders of acetylcholine and serotonin transmission, which are caused by diseases, medication or drug use, are often accompanied by hallucinations, which can also include bodily illusions, e.g. the feeling of falling into an abyss (Epstein et al., 2002).

These no-goes may in part be due to language problems and make it indispensable that the authors should ask a native English speaking person to correct their language. The nontechnical style of the essay is welcome, but expressions are at times difficult to understand (twice on p. 4 «this group of forms»), especially when nuances in experiential or phenomenal descriptions are important for grasping an intended classification, categorization or comparison with related bodily misperceptions.  

For the last of the last of the very last version of this manuscript we can ask a native speaker for correction. But I think it may be to early to do this now, because I’m afraid of more critics.

Reviewer 2 Report

Thank you for inviting me to review this manuscript. It addresses an interesting topic such as coenaesthesia through different case-studies.

It’s a well and pleasant written manuscript, and it’s suitable for publication.

However there are some changes that should be addressed.

In some points it lacks of references. For instance, in the introduction section just two references have been presented. Since in this section there is much theorethical information, I recommend using much more recent references.

Other minor changes:

-      -   Introduction: I suggest beginning the section with a more general paragraph presenting the main topic and the first paragraph of the female patient could better fit elsewhere of the introduction ahead.

-        - Regarding the same first paragraph, line 1, I would recommend changing “an epileptic” for “One of our female patients who suffer from epilepsia” instead.

-        - Section 2, Body hallucinations in mental disorders, line 2: I would recommend deleting the sentence: “(i.e. one of the authors)”. It reduces the scientific profile of the manuscript.

Author Response

REVIEW #2

Thank you for inviting me to review this manuscript. It addresses an interesting topic such as coenaesthesia through different case-studies.

It’s a well and pleasant written manuscript, and it’s suitable for publication.

However there are some changes that should be addressed.

In some points it lacks of references. For instance, in the introduction section just two references have been presented. Since in this section there is much theorethical information, I recommend using much more recent references.

Literature sources are doubled now (30 instead of 14).

Other minor changes:

-      -   Introduction: I suggest beginning the section with a more general paragraph presenting the main topic and the first paragraph of the female patient could better fit elsewhere of the introduction ahead.

As early as 1911 Head and Holmes (1911) were among the first to attempt to understand how the brain maps the human body. They developed one scheme for the passive perception of stimuli on the skin (“superficial scheme”) and another for the position and movement of limbs (“posural scheme”). Most contemporary authors use terms like "body schema", "body image" or "body representation". In 1979 Critchley used the term "corporeal awareness". Here he tried to integrate an affective and emotional component in addition to the perceptual and conceptual component. As Berlucci & Aglioti noted in 2010, there is still no universally accepted terminology. The present article will therefore not deal intensively with the problem of definitions, this narrative review is more about the fact that there is a large number of disorders that lead to a confusion of the mental picture that our brain draws of its own body. Since there are countless types of body image disorders, the article is limited to selective examples of the most interesting and sometimes obscure deviations. Another conceivable distinction concerns motor and sensory disorders. This article deals primarily with changes in the perception of the body, motor disorders are only marginally discussed. We will start with a clinical single case study:

-        - Regarding the same first paragraph, line 1, I would recommend changing “an epileptic” for “One of our female patients who suffer from epilepsia” instead.

One of our female patients, who suffer from epilepsia and had a brain haemorrhage, described how she once had the feeling that ….

-        - Section 2, Body hallucinations in mental disorders, line 2: I would recommend deleting the sentence: “(i.e. one of the authors)”. It reduces the scientific profile of the manuscript.

In the run-up to an illness, when you don't quite feel whether you're still healthy or already sick, many people occasionally experience strange body changes where, for example, a hand suddenly feels huge and heavy as lead. Physical misperceptions caused by fever, up to and including fever hallucinations, are known here. Fever is induced when pro-inflammatory cytokines trigger prostaglandin E2 synthesis by binding to receptors on brain endothelial cells (Blomquist & Engblom, 2018). Fever affects several neurotransmitters, Cox and Lee specifically listed: norepinephrine, 5-hydroxytryptamine, and acetylcholine, which in turn can severely disrupt the body's recognition.

Reviewer 3 Report

Review for Healthcare

How the mind creates the bod – and what can go wrong: Case-studies of Coenaesthesia

Thank you for the opportunity to review this manuscript. While the manuscript is interesting, in its current form it lacks critical analysis and is very descriptive. There are several very broad issues that would need to be addressed before the manuscript could be considered for publication:

·       There are many instances throughout the manuscript where long tracts of text are not accompanied y any supporting references. Statements are mad as if they are fact, but no citations are given. Lines 69-82, lines 291-295, lines 321-325, lines 392-398 are but some examples in the manuscript where this has occurred.

·       Lines 361-368 need amending. Although Moody described a ‘prototypical’ NDE in his work that occurred in a specific sequence, the features of NDEs do not per se occur in a specific sequence. Each NDE is unique, some have many features while others have few, and there appear to be cross-cultural differences in the phenomenology of features. There are numerous references available to support what I have said here. I suggest the author/s amend the statements to ensure they are not mis-leading.  

·       Overall, the manuscript is very descriptive and does not provide a lot of nuance or critical reflection on the material that it covers. It seems that all experiences where body perception is altered are brought together under the umbrella of coenaesthesia (as the title suggests), which are then likened to ‘bodily hallucinations’. There is no discussion on how they might be differentiated from each other or some of the proposed mechanisms that underly each (and therefore might provide some information about how they are differentiated). There is also no real definition of what ‘bodily hallucinations’ are, and the use of the terms implies pathology. Many who have had an NDE and researchers in the area alike, would argue that NDEs, as an example, are not bodily hallucinations. There is literature to support this. Perhaps the author/s may need to refine the manuscript in light of how the term coenaesthesia applies to each of the experiences mentioned.

·       The manuscript seems to conclude and imply that ‘nerve connections’ are responsible for changes in body awareness. My understanding is that the established evidence points to correlation rather than causation, so perhaps this may also be useful to state.

Author Response

Thank you for the opportunity to review this manuscript. While the manuscript is interesting, in its current form it lacks critical analysis and is very descriptive. There are several very broad issues that would need to be addressed before the manuscript could be considered for publication:

  • There are many instances throughout the manuscript where long tracts of text are not accompanied y any supporting references. Statements are mad as if they are fact, but no citations are given. Lines 69-82, lines 291-295, lines 321-325, lines 392-398 are but some examples in the manuscript where this has occurred.

Literature sources are doubled now (29 instead of 14).

  • Lines 361-368 need amending. Although Moody described a ‘prototypical’ NDE in his work that occurred in a specific sequence, the features of NDEs do not per se occur in a specific sequence. Each NDE is unique, some have many features while others have few, and there appear to be cross-cultural differences in the phenomenology of features. There are numerous references available to support what I have said here. I suggest the author/s amend the statements to ensure they are not mis-leading.  

Moody's paper described prototypical near-death experiences with a specific order; However, more recent studies show that this row sequence occurs only extremely rarely. Most of those affected report only individual parts of these experiences, only rarely was the whole process experienced and then often not in the sequence prescribed by Moody. There are also cultural differences. It is scientifically disputed whether one really looks over the border at the end of life or whether it is a question of hallucinations as a result of a neuronal catastrophe reaction (Kasten & Geier, 2014).

  • Overall, the manuscript is very descriptive and does not provide a lot of nuance or critical reflection on the material that it covers. It seems that all experiences where body perception is altered are brought together under the umbrella of coenaesthesia (as the title suggests), which are then likened to ‘bodily hallucinations’. There is no discussion on how they might be differentiated from each other or some of the proposed mechanisms that underly each (and therefore might provide some information about how they are differentiated). There is also no real definition of what ‘bodily hallucinations’ are, and the use of the terms implies pathology. Many who have had an NDE and researchers in the area alike, would argue that NDEs, as an example, are not bodily hallucinations. There is literature to support this. Perhaps the author/s may need to refine the manuscript in light of how the term coenaesthesia applies to each of the experiences mentioned.

Every morning when we wake up, the brain forms an image of our own body. One knows that the right foot is part of the body and puts the stocking over it. When you get up and walk out of the bedroom, you can do this because a complex network of neuronal assemblies in your head work together. The sensitive nerve fibres play an important role here, constantly reporting which posture each individual limb is currently in; the organ of equilibrium behind the ears tells us which position we are currently in and the eyes support us here with a visual orientation in the entire room. In the CNS, it is first the somotosensory cortex in the parietal lobe that processes the input from e.g. Merkel cells, Meissner, Ruffini and Vater-Paccini corpuscles, as well as from the nociceptors, and gives us a sense of what our body is doing. The cerebellum plays a crucial role in keeping the body in a stable position, the motor cortex in the frontal lobe then controls movements adapted to the situation. The pre-motor cortex and the supplementary motor cortex plan complex movements, but also contains highly trained sequences of movements that can then be largely automated, such as riding a bicycle, swimming or playing a musical instrument. Both areas are on the posterior frontal lobe; the supplementary motor area on the upper frontal lobe is used for the preparation and execution of actions, i.e. the planning and selection of learned, non-stimulus-induced movements and speaking. It triggers voluntary motor activity and is located in front of the premotor cortex, which is used for sensorimotor integration and the preparation and execution of voluntary movements. The network also includes an area in the transitional region between the temporal and parietal lobes, which has a superordinate hierarchical task and tells us what actually belongs to our own body.

This neuronal network can suffer disturbances, for example through drugs or temporary circulatory disturbances. But it can also suffer permanent damage, e.g. in the case of strokes, craniocerebral traumas or other neurological disorders such as polyneuropathy. One can basically distinguish between two forms of such disorders of bodily sensation. On the one hand, there is the (1) negative variant, in which the affected person feels too little of his or her body, such as the insensitivity of one half of the body in many patients with paralysis after a stroke. In the (2) positive variant, on the other hand, the affected person feels sensory phenomena that are not present at all. Depending on the symptom, these are called e.g. tactile hallucinations, paraesthesia and coenaesthesia.

In the case of paraesthesia and tactile hallucinations, the affected person feels, for example, that thousands of insects are crawling over them, that worms are boring through their skin, that they are being pushed from behind or that strangers are lying on top of them and crushing them. In alcohol withdrawal delirium, there is often a phase in which sufferers have such tactile hallucinations. Möller (1975) reported of a schizophrenic who had the feeling at night that her body was being sawn through and reassembled. However, she only felt pain when the (non-existent) cuts were touched. Another patient, a 50-year-old Spaniard, suffered from the tactile hallucination that someone was blowing air into his stomach, as if through a straw, and that his stomach was getting fatter as a result.     

Coenaesthesia and bodily hallucinations are also perceptual disorders. According to Fuchs (1995), the term coenesthesia is composed of the two Greek words 'koiné' and 'aísthesis', and originally meant "general sensation", i.e. the perception of one's own body condition. As early as 1811, Reil listed a number of disorders under the term “Gemeingefühl” (see Fig. 1), which gave the soul distorted ideas about the body. Even if these diseases two centuries ago used to have different terms, he probably meant, for example, bulimia, nymphomania, cravings, delusional beliefs about bodily dysfunction or metamorphosis, hysteria and hypochondria. As a result, in the last 200 years coenesthesia has become a component of many mental disorders. Today the term coenesthesia is found almost exclusively in pathological contexts, for example as an abnormal or bizarre bodily sensation in schizophrenia, while the recognition of one’s own body is divided up in interoception, exteriorperception and proprioperception (see below).

Coenaethesia related to bizarre bodily misperceptions and body hallucinations overlap and are sometimes difficult to distinguish. When people hear the term "hallucinations," many only think of hearing voices or seeing objects or scenes that are not real. However, hallucinations can occur in all sensory modalities; this also includes tactile physical misperceptions.

These kinds of bodily sensations are often difficult to describe, so that many sufferers feel compelled to use grotesque comparisons. These include, for example, the feeling that individual body parts are suddenly heavy as lead or huge. Albert Hofmann, the inventor of LSD, described such changes in the body schema. Other symptoms can be that the body or individual body parts grow, become thicker, heavier or lighter. Sometimes those affected also have the feeling that they are made of stone, metal, wood or plastic on the inside. Still other patients suffer from the feeling that their liver is rotting, they feel the heart being cut out, the intestines decomposing, the spleen being parasitised, the pancreas decomposing, the lungs being eaten away or the brain liquefying. Sometimes there are also hallucinations of movement, in which the person concerned has the feeling that his body parts are performing movements independently, for which he himself is not at all responsible. In addition, there are pulling and pressure sensations inside the body, even the sensation of being strangled.

Vestibular and kinaesthetic hallucinations change the perception of the position of one's own body in space. These include, for example, the feeling that everything is spinning, that one is falling endlessly, floating (levitation) or suddenly becoming heavier and heavier and sinking into the bed or the ground.

Such hallucinations can occur with lesions of the upper midbrain and adjacent thalamus (Epstein et al., 2002). In waking mode, the thalamus faithfully relays sensory input to the cortex; in sleep mode, or due to disturbances of the brain-function it does not fulfill this task completely. This change of body-recognition involves several neurotransmitters, particularly acetylcholine and serotonin, which are involved in selective attention and cortical processing. Disorders of acetylcholine and serotonin transmission, which are caused by diseases, medication or drug use, are often accompanied by hallucinations, which can also include bodily illusions, e.g. the feeling of falling into an abyss (Epstein et al., 2002).

  • The manuscript seems to conclude and imply that ‘nerve connections’ are responsible for changes in body awareness. My understanding is that the established evidence points to correlation rather than causation, so perhaps this may also be useful to state.

Even if we now understand the nerve connections in neural networks better, it remains a mystery how exactly the knowledge about our body image arises. As Gabourey Sidibe in an internet page says: “It doesn’t have anything to do with how the world perceives you. What matters is what you see.”

Round 2

Reviewer 1 Report

Much improved version, thank you also for considering (and commenting on) the suggested changes.

Acceptable as a review.

English still not corrected by a nativ speaker; also check some typos (e.g., "postural").

Reviewer 3 Report

My initial comments appear to have been addressed.